



# Aerosol hygroscopicity and its link to chemical composition in coastal atmosphere of Mace Head: marine and continental air masses

Wei Xu[1,2], Jurgita Ovadnevaite[1], Kirsten N. Fossum[1], Chunshui Lin[2], Ru-Jin Huang[2,3], Colin O'Dowd[1], Darius Ceburnis[1]

[1]School of Physics, Ryan Institute's Centre for Climate & Air Pollution Studies, and Marine Renewable Energy Ireland, National University of Ireland Galway, University Road, H91CF50 Galway, Ireland

[2]State Key Laboratory of Loess and Quaternary Geology, Center for Excellence in Quaternary Science and Global Change, and Key Laboratory of Aerosol Chemistry and Physics, Institute of Earth Environment, Chinese Academy of Sciences, 710061

Xi'an, China

[3]Open Studio for Oceanic-Continental Climate and Environment Changes, Pilot National Laboratory for Marine Science and Technology (Qingdao), 266061Qingdao, China

*Correspondence to*: Ru-Jin Huang (rujin.huang@ieecas.cn) and Colin O'Dowd (colin.odowd@nuigalway.ie)

**Abstract.** Chemical composition and hygroscopicity closure of marine aerosol in high time resolution has not been yet achieved because of the difficulty in measuring refractory sea-salt concentration in near-real time. In this study, attempts were made to achieve a closure for marine aerosol based on a humidified tandem differential mobility analyser (HTDMA) and a high-resolution time-of-flight aerosol mass spectrometer (AMS) for wintertime aerosol at Mace Head, Ireland. The aerosol hygroscopicity was examined as a growth factor (GF) at 90 % relative humidity (RH). The corresponding GFs of 35, 50, 75,

110 and 165 nm particles were $1.54 \pm 0.26$, $1.60 \pm 0.29$, $1.66 \pm 0.31$, $1.72 \pm 0.29$ and $1.78 \pm 0.30$ (mean ± standard deviation), respectively. Two contrasting air masses (continental and marine) were selected to study the temporal variation in hygroscopicity and the results demonstrated a clear diurnal pattern in continental air masses, while no diurnal pattern was found in marine air masses. In addition, the winter time aerosol was observed to be largely externally mixed in both contrasting air masses. Concurrent high time resolution $PM_1$ (particulate matter < 1 μm) chemical composition by combined AMS and

MAAP measurements comprising of organic matter, non-sea-salt sulphate, nitrate, ammonium, sea-salt and black carbon (BC) were used in predicting aerosol hygroscopicity using the Zdanovskii–Stokes–Robinson (ZSR) mixing rule. A generally good agreement ($r^2 = 0.824$, slope = 1.02) was found between HTDMA measured growth factor (GF_HDTMA) of 165 nm particles and AMS+MAAP bulk chemical composition derived growth factor (GF_AMS). Over 95% of the estimated GF exhibited less than 10% deviation for the whole dataset and the deviation was mostly attributed to the neglected mixing state as a result of

bulk PM1 composition.



## 1. Introduction

Marine aerosol is probably the most important component of natural aerosol in terms of climate effect (O'Dowd and de Leeuw,
2007), because over 70% of the Earth's surface is covered by world oceans. There are two ways that marine aerosol can exert
its impact on global climate, one by scattering the incoming solar radiation, and the other by acting as cloud condensation
nuclei (CCN). Hygroscopicity - the ability of water vapor uptake by aerosol, plays a significant role for both. Hygroscopicity
affects the mass of aerosols by increased aerosol liquid water content and enhances particle light scattering, therefore, cooling
the atmosphere directly. Furthermore, hygroscopicity has a large impact on CCN activation and cloud droplet formation,
modifying cloud radiative forcing and the hydrological cycle (Twomey, 1974; 1977).

Aerosol hygroscopicity is determined by its chemical composition. Closure studies which attempted to predict hygroscopicity
based on chemical composition measurements have improved understanding of the relationship between aerosol
hygroscopicity and chemical composition in various environments. Thanks to the wide use of aerosol mass spectrometry
(AMS), chemical composition of aerosols is now available to attempt closure with hygroscopicity data in a high time resolution.
The collocated aerosol chemistry and hygroscopicity measurements have been of great importance in reconciling sub-saturated
particle hygroscopicity with its chemical composition thereby identifying knowledge gaps. However, it is widely accepted that
sea-salt (the main component of marine aerosol) measurements by AMS are challenging because of its semi-refractory nature
resulting in incomplete chemical composition and unrealistic hygroscopicity. The hygroscopicity of marine aerosol has been
intensively studied, including studies in the Arctic (Zhou et al., 2001), Atlantic (Swietlicki et al., 2000) and Pacific (Berg et
al., 1998) oceans, but chemical composition and hygroscopicity closure studies are still very limited. A hygroscopicity and
chemical composition study conducted in the Northeastern Pacific (Kaku et al., 2006) found over 30% overestimation of
growth factor (GF) by using the Zdanovskii–Stokes–Robinson (ZSR) mixing rule. The study speculated that the overestimation
was caused by the non-ideal behavior of organics. Investigation into the hygroscopicity of aerosol in Antarctica by using an
impactor for size-segregated composition of marine aerosol particles, found the hygroscopicity mainly driven by inorganic
salts (Asmi et al., 2010). However, due to the limitation of sampling technique (filters and impactors) and a short sampling
period, they were unable to capture the temporal evolution of chemical composition, hindering the detailed analysis of real
time linkage to hygroscopicity.

This study aimed at characterizing marine and continental aerosol during winter period of low marine biological productivity
at the coastal Mace Head Atmospheric Research Station situated at the boundary of the North East Atlantic and rural West of
Ireland. The aerosol hygroscopicity was measured in-situ in sub-saturated conditions (RH 90%) using a humidified tandem
differential mobility analyzer (HTDMA). Aerosol hygroscopicity parameter was also estimated using chemical composition
data using near real-time chemical composition measurements including sea-salt by a high-resolution time-of-flight aerosol
mass spectrometer (HR-ToF-AMS) and Multiple Angle Absorption Photometer (MAAP). Two contrasting cases were
analyzed in detail to represent continental and marine air masses in which the hygroscopic aerosol properties were expected





to differ greatly. To the best of our knowledge, this is the first closure study on aerosol hygroscopicity and chemical composition including sea salt for marine aerosols in high time resolution.

## 2. Method

### 2.1 Site Description

The Mace Head Atmospheric Research Station (MHD) is located on the North Atlantic coast of Ireland, co. Galway at 53°19′36″N, 9°54′14″W (O'Connor et al., 2008). Air is sampled from the main community sampling duct which draws air from 10 m above ground level and is positioned 80-120 m from the ocean depending on the tide. Meteorological data is recorded at the station, including rainfall, solar radiation, wind speed, wind direction, temperature, RH, and pressure (available at www.macehead.org). The measurements were conducted from 1st Jan to 23rd March 2009 and comprised of 1300 hours of

valid HTDMA and AMS data. Air masses were tracked using HYSPLIT (Rolph et al., 2017) 72 h backward trajectories with an endpoint of 500 m above mean sea level at Mace Head with Global Data Assimilation System (https://www.ncdc.noaa.gov/data-access/model-data/model-datasets/global-data-assimilation-system-gdas).

### 2.2 Instrumentation

### 2.2.1 HTDMA

The hygroscopic growth factor of aerosol particles was measured with an HTDMA (Liu et al., 1978; Rader and McMurry, 1986; Swietlicki et al., 2008). The HTDMA at Mace Head (MHD) which was described in great detail in previous studies (Bialek et al., 2012; 2014) consisted of a dry Hauke-type differential mobility analyzer (DMA, RH<10%, dried by a Nafion™ dryer), Gore-Tex™ humidifier, second Hauke-type DMA and a condensation particle counter (CPC, TSI model 3772). To stabilize the RH the second DMA was placed in a temperature-controlled box. Four Rotronic RH/temperature sensors and an

Edgetech Dewmaster dewpoint chilled mirror sensor were used to monitor the RH fluctuation within the system and the humidifier was controlled by an analogue to digital and digital to analogue feedback system. The first DMA was used to select monodisperse particles with a certain electrical mobility. The monodisperse particles were then humidified and hygroscopicity growth probability distribution function is then produced by the second DMA and the CPC. Since the dry diameter of aerosol is well established, the hygroscopic growth factor can be calculated by measuring the aerosol size distribution at selected RH.

To retrieve growth factor from raw data, and to correct the broadening of DMA distribution, a piece linear inversion algorithm was used (Gysel et al., 2009). In this study, the first DMA was constantly at an RH 10%, while the second DMA was set at RH=90%. The dry particle diameters selected by the first (dry) DMA were 35, 50, 75, 110 and 165 nm, with a scan duration of 180 seconds such that the full cycle through all diameters took 15 minutes. The sample and sheath flow rates were 1 L min⁻¹ and 9 L min⁻¹, respectively. The operation and quality assurance procedure followed standard configuration and deployment

recommended by the European Supersites for Atmospheric Aerosol Research (EUSAAR) network project (Duplissy et al., 2009).



### 2.2.2 Chemical composition (HR-ToF-AMS and MAAP)

The chemical composition was measured using HR-ToF-AMS (Aerodyne Research Inc. Billerica, MA) (DeCarlo et al., 2006) which has a vacuum aerodynamic cut-off diameter of 1 μm. Regular calibrations with ammonium nitrate were performed and

the composition-dependent collection efficiency was applied. The AMS provided the mass concentration of organic matter, ammonium, non-sea-salt sulfate, nitrate, methanesulfonic acid (MSA) and sea-salt which was retrieved by using a $^{23}Na^{35}Cl^+$ ion signal at m/z 58 and a scaling factor of 51 (Ovadnevaite et al., 2012). The operational details of the HR-ToF-AMS are described in Ovadnevaite et al. (2014). The degree of neutralization (DON) of bulk aerosol was calculated as: DON = $\frac{n(NH_4^+)}{2nSO_4^{2-}+nNO_3^-}$. The concentration of optically absorbing black carbon (BC) was measured by a Multi-Angle Absorption

Photometer (MAAP, Thermo Fisher Scientific model 5012). The MAAP operated at a flow rate of 10 L min$^{-1}$ at 5 min time resolution. The MAAP measured transmittance and reflectance of BC contained particles at two angles to calculate optical absorbance, as described in Petzold and Schönlinner (2004).

### 2.3 Hygroscopicity data analysis

The growth factor (GF) of aerosol particles undergoing humidification was obtained by: $GF = \frac{D}{D_0}$, where D and $D_0$ are the electrical mobility diameters of humidified and dry aerosols, respectively.

One of the HTDMA features is the ability to reveal aerosol mixing state by detecting the presence of more than one particle growth mode. Each growth mode represents different water uptake properties, indicating a different chemical composition of each mode. The growth factor probability distribution function was separated into four growth modes according to their GF

range: Near Hydrophobic mode (NH, 1< GF< 1.11), Less Hygroscopic mode (LH, 1.11< GF< 1.33), More Hygroscopic mode (MH, 1.33< GF< 1.85) and Sea Salt mode (SS, 1.85< GF). For particles in a specific GF range, for example LH, the number fraction of this mode (nf_LH) was derived from the retrieved probability density function c(GF, D) as: nf_LH = $\int_{1.11}^{1.33} c(GF, Do)gGF$. It should be noted that this categorization was not always representative. For example, in highly acidic marine aerosol, the MH mode peaked at 1.8< GF< 2.0 resulting from high highly hygroscopic H$_2$SO$_4$ with GF up to 1.9 rather

than sea salt. Due to the spread of GF-PDF, some sections of PDF representing non-neutralized particles could be categorized into SS mode, which would then result in underestimation of averaged GF for both MH and SS modes, and overestimation of the number fraction of SS mode.

### 2.4 Hygroscopicity-Chemistry Closure

The mass concentrations were converted to volume fractions of individual components (organics, NH$_4$HSO$_4$, H$_2$SO$_4$, NH$_4$NO$_3$, (NH$_4$)$_2$SO$_4$, MSA, sea-salt and BC) by using a simplified ion pairing scheme (Gysel et al., 2007). The GFs values of individual components are summarized in Table 1. Although the hygroscopicity of inorganic compounds is well understood and established, it is still challenging to quantify the hygroscopicity of organic matter ranging from 1 to 1.5 or from hydrocarbon to oxalic acids (Kreidenweis and Asa-Awuku, 2014), however, most of the anthropogenic organics have GF <1.2. In this study,





we first used a fixed GF value of 1.18 for organics which was the averaged value from several closure studies (Wang et al.,
        2018; Yeung et al., 2014) and a constant density of 1400 kg m$^{-3}$ was used.

        Assuming a constant GF and density value for organics may induce a bias for closure studies because the hygroscopicity of
        organics differs according to their molecular structure, air mass history, or oxidation level. The GF$_{MSA}$=1.71 was calculated by
        kappa value which in turn was obtained by AIOMFAC model (Fossum et al., 2018; Zuend et al., 2011). The hygroscopicity

of inorganic sea-salt was found to be 8-15% lower than that of pure NaCl, therefore, a GF value 2.22 of inorganic sea-salt (at
        RH 90%) was used (Zieger et al., 2017). The closure between the measured and predicted values was characterized by a linear
        regression and the corresponding values of R$^2$ (the variance, which is a square of the correlation coefficient) and regression
        slope.

        Growth factor estimation was based on the Zdanovskii-Stokes-Robinson (ZSR) mixing rule (Stokes and Robinson, 1966) using

measured aerosol chemical composition, which assumes that water uptake of the mixture is equivalent to the sum of the water
        uptake of individual substances. GF calculated from the bulk chemistry of the HR-ToF-AMS(GF_AMS) can be written as:
        $GF\_AMS = \sum_i v_i GF_i$, where $v_i$ is the volume fraction of the compound in the dry particle, and $GF_i$ is the growth factor of the
        individual chemical components. In the above equation, any interaction between the solutes is neglected and the volume of the
        dry mixture is the sum of the volumes of its dry components.


## 3. Result and Discussion

### 3.1 Meteorology and air mass origin

        The measurement period spanned from the 1$^{st}$ January 2009 to the 23$^{rd}$ March 2009. Data including meteorological parameters
        and aerosol chemical composition are shown in Fig 1. The average ambient temperature and RH for the entire period were 6.5

± 2.5 °C and 85.8 ± 8.8 %, respectively.

        The measurement period was examined in terms of contrasting air mass origin and two continental events (C1, C2) and two
        marine events (M1, M2) were selected. These contrasting events are highlighted in the time series shown in Fig. 1 and were
        expected to reveal greatly different hygroscopic properties of aerosol particles. The air mass backward trajectories for these
        four events are shown in Fig. 2. Events C1 and C2 represented air masses which originated over continental Europe 72 hours

prior to being transported across the UK and Ireland towards Mace Head. Events M1 and M2 were considered representing
        clean marine air originating over the North East Atlantic Ocean and transported to the west coast of Ireland. The start and end
        time of each event is summarized in Table S1. The mass concentrations of chemical composition, including non-sea-salt sulfate,
        nitrate, ammonium, organic, MSA, sea-salt and black carbon of each event are summarized in Table S2. It is important to
        emphasize that marine air masses are not always pristinely clean despite of advecting over the oceanic waters. Therefore, only

data with BC <15 ng m$^{-3}$ and wind direction within the 190 to 280° sector were included in data analysis of marine events and
        subsequently summarised in Table S2.The mean BC concentrations during M1 and M2 events were 10.1 and 9.9 ng m$^{-3}$,



respectively, demonstrating the value of conservative approach to qualifying pristine marine air masses and the corresponding data capture is presented in Figure S1 for M1 event

## 3.2 Aerosol hygroscopicity

3.2.1 Overview of hygroscopicity measurements

During the full winter measurement period, aerosols displayed temporal variation in GF-PDF across all sizes, and overall the larger particles clearly exhibited larger GF values (Fig. 3). The mean GF of 35, 50, 75, 110 and 165 nm dry mobility diameter particles were $1.54 \pm 0.26$, $1.60 \pm 0.29$, $1.66 \pm 0.31$, $1.72 \pm 0.29$, $1.78 \pm 0.30$, respectively.

The GF-PDFs were observed to be highly size dependent throughout the sampling period (Fig. 3) and, in addition, different modal pattern (single mode, bimodal and/or trimodal) were found for all measured particle sizes but of different frequency of occurrence. The occurrence of single mode profiles increased with decreasing $D_0$ size, such as the frequency of occurrence was 10.4 % for 35 nm particle diameter, 8.8 % for 50 nm, 7.6 % for 75 nm, 4.7 % for 110 nm and 3.0 % for 165 nm dry particle diameter. A few trimodal patterns were observed, particularly in marine air masses, and the occurrence of trimodal profiles similarly increased with decreasing size, such as the frequency of occurrence was 9.8% for 35 nm, 7.9 % for 50 nm, 6.5% for 75 nm, 3.8% for 110 nm and 1.9% for 165 nm dry particle diameter. Overall, bimodal GF-PDF profiles dominated the whole winter period regardless of size (Fig. 3), suggesting that the sampled aerosol was largely externally mixed at Mace Head during the entire winter season. To determine the influence of air mass, we examined the hygroscopicity and chemical composition of marine and continental aerosol in the following sections.

## 3.2.2 Continental air masses

No precipitation was observed during continental air mass events, and measured temperatures were typical of Mace Head winter seasons, ranging from 1 to 6 °C, while RH ranged from 70 to 100%. Wind speed peaked at a maximum of 17 m s$^{-1}$ and a minimum of below 5 m s$^{-1}$.

Figure 3 provides an overview of the GF-PDFs and average GFs of pre-selected aerosol particles. Throughout the C1 and C2 events, particles with a dry size $D_0 > 75$ nm exhibited bimodal or trimodal GF-PDFs with a mode of more-hygroscopic and a mode of less hygroscopic or a mode of near hydrophobic nature. Particles with sizes $D_0 < 50$ nm were rather different and dominated by LH mode and NH modes. Completely non-hygroscopic particles (GF~1) were not observed, but some of the GF-PDF data spread reached NH mode, indicating some extent of internal mixing, however, dominant multimodal pattern clearly demonstrated mostly external mixing. External mixing has also been observed in other studies in winter, especially in locations with large anthropogenic influence (Swietlicki et al., 2008).

Figure 4 depicts the number fraction of each growth mode type by measured particle size over the winter measurement period. The MH mode was dominated by 165 nm particles, while the LH mode and NH mode became more prominent as the size decreased. A similar size dependence mode distribution was also observed in Beijing (Wu et al., 2016) and the southern Sweden (Fors et al., 2011). The number fraction of NH (nf_NH) were similar for all sizes, nf_LH decreased with increasing





particle size nf_MH mode increased with increasing size. It is generally argued that larger particles have typically undergone atmospheric aging and cloud processing (such as coagulation, droplet coalescence, condensation of semi-volatile gases, chemical reactions and photo-oxidization) for a longer period of time thus acquiring additional mass, growing in size and exhibiting more hygroscopic features. The SS mode was very small in continental air masses as expected (frequency of occurrence of 1 % during C1 and 5 % during C2), but externally mixed sea-salt was clearly discerned nevertheless. As shown

in Fig. S2, the average GF of 110 nm and 165 nm particles showed a clear diurnal pattern, which peaked at about 11:00 every day and reached a minimum at 20:00. This trend was similar to the GF observation in Po Valley (Bialek et al., 2014) or Oklahoma site (Mahish and Collins, 2017) and is often attributed to a shallower and more stagnant boundary layer during the night with temperature inversions arising from radiative cooling of the surface.

   The AMS and MAAP measurements are shown in Fig. 1(top), and the mean ± standard deviation of total mass concentration,

BC, organic matter, nitrate, ammonium, and non-sea-salt sulfate are shown in Table S2. The mass concentrations of BC, during C1 and C2 were 500 and 518 ng m$^{-3}$, and nitrate mass loadings were 0.92 and 4.06 μg m$^{-3}$), suggesting a heavy anthropogenic impact during continental events. The mass concentrations of sea-salt (0.17 and 0.13 μg m$^{-3}$) and MSA (0.007 and 0.006 μg m$^{-3}$) were relatively low, suggesting little impact of marine sources during C1 and C2 during winter.

**3.2.3 Marine air masses**

   Aerosol hygroscopicity in the marine air masses was dramatically different from the continental air masses. The meteorological conditions and chemical composition of each event are shown in Fig. 1. During marine air mass events M1 and M2, the wind speed varied from 4 to 20 m s$^{-1}$, wind direction varied from 180- 320° and corresponded to clean maritime conditions at MHD (O'Dowd et al. 2014). The RH and temperature ranged from 70-100 % and 0-10 °C, respectively. The marine GF-PDF of M1

and M2 were mostly bimodal, indicating that the aerosol was mostly externally mixed, but generally much more hygroscopic than during continental events. The mean GF of M1 and M2 are shown in Table S3, which ranged from 1.8 to 2.1, suggesting the highly hygroscopic nature of marine aerosol. The diurnal pattern of M1&M2 is shown in Figure S1 and contrary to C1& C2 air masses the M1&M2 did not revealed a clear diurnal pattern, which was likely due to well mixed marine boundary layer and stable temperature over the ocean. Long term data is required to form a solid conclusion which will be the scope of further

research.

   The MH mode was ubiquitous in wintertime sampled marine aerosol (observed in all measured particle sizes), and accompanied by SS mode, and some of LH or NH modes in the smallest particle sizes. Quite interestingly, a significant number fraction of sea-salt could be detected down to particle diameters of 35 nm in accordance to nano particle modes in sea spray source function developed by Ovadnevaite et al. (2014). In this study the greatest number fraction of the SS mode was observed

around a particle diameter of 75 nm again in line with the aforementioned sea spray source function. The NH and LH were more pronounced in the smaller sizes. Applying a pristine marine criteria of BC <15 ng m$^{-3}$ and wind direction from 190 to 300°, the NH and LH modes were significantly reduced across all sizes and effectively absent in the largest particle sizes (Fig. S1), but the NH and LH mode of 35 nm still existed. Given the conservative BC threshold and the absence of low





hygroscopicity modes in larger particles local anthropogenic contamination can be excluded, but the true origin of less
hygroscopic nano particles will the subject of further research. The averaged GF increased with increasing size for MH modes
(e.g. the averaged GF of the MH mode increased from 1.52 for 35 nm to 1.70 for 165 nm particles). The average GF of the
MH mode in 110 nm GF-PDF was around 1.8. This is similar to the GF of ammonium bisulfate (GF=1.79), indicating that
aerosol in the MH mode was non-neutralized sulfate originating from marine DMS oxidation and lack of ammonia and
resulting in largely acidic particles. Moreover, the marine GF of the MH mode was higher than that of the continental event,
which could be attributed to the difference in DON. The DON for C1, C2, M1, M2 were 0.88, 0.93, 0.24, 0.03, respectively,
clearly suggesting a higher contribution of sulfuric acid and ammonium bisulfate in marine air masses. In contrast to previous
studies at coastal sites of Hong Kong during winter seasons reporting very low frequency of occurrence of the SS mode (Yeung
et al., 2014), our observations indicated a large presence of SS mode during wintertime as a result of long advection over the
stormy North Atlantic.

The comparison of GF between continental and marine events is shown in Fig. 5 where GFs increase with aerosol size in both
continental and marine events, but the size dependence is rather different. The difference between GFs of 35 and 50 nm are
smaller in continental events than that of marine events. On the contrary, the difference among 75, 110 and 165 nm is smaller
in marine events.

### 3.3 Chemical composition closure study


The size resolved GFs measured by the HTMDA (denoted as GF_HTDMA) were plotted against GFs estimated by the ZSR
mixing rule using AMS chemical composition data (denoted as GF_AMS). The linear regression was used to fit the GF_AMS
and GF_HTDMA, the slope of a nonzero-intercept linear regression fit reflecting how well the estimation agreed with the
measurements. As shown in Fig. 6, the regression slopes were 0.91 and 1 for 35 and 165 nm, respectively, and the variance
increased from 0.61 to 0.84 with the increase of dry diameter $D_0$, suggesting the closure agreement getting better for larger
sizes. For example, the GF_AMS of 35 nm $D_0$ aerosols showed an overestimation with over 93% of the data points located
outside of the 10% deviation from the 1:1 line. The comparison of 165 nm aerosols to bulk chemistry was very good with over
95% of data points lying within 10% deviation of the 1:1 line. The slope of linear regression is 1.02 suggesting there was no
systematic error in GF estimation. The results of the comparison were suggesting that (1) the chemical composition which was
used in deriving GF was bulk PM1 data which may differ significantly from Aitken mode particles, but approximating to
accumulation mode particles thus affecting poorer comparison of Aitken mode particles and (2) the Kelvin effect, which would
become significant at small sizes was neglected in calculation. Contrary to the study of Hong et al. 2018, the correlations
between GF_AMS and GF_HTDMA of even small sizes were correlating much better than no-correlation found by (Hong et
al., 2018).

Given the best obtained agreement for the larger sizes, and considering the relevance to the cloud condensation nuclei, we now
focus on the closure results of 75, 110 and 165 nm particles to assess event results. As shown in Fig. 6, although a general



closure agreement was good, large amount of data points was still scattered around the regression line. Comparison between GF_AMS and GF_HTDMA was plotted for continental and marine events, as shown in Fig. 7a) and Fig. 7b).

The regression lines were approaching the 1:1 line with the increasing size, for example, the $R^2$ were 0.47 and 0.18 of 75 nm
continental and marine events. For 165 nm $D_0$ aerosol, the regression coefficients were 0.72 and 0.54, and the slopes were 1.1 and 0.85 for both continental and marine events.

The regression results were reasonable even for 75nm size, but were certainly getting better with the increasing size for the continental events. However, the improvement was not as significant for marine events (Figure 7). Although a few data points were outside of the 10% deviation range of 75 nm, the $R^2$ of 75 nm as low as 0.18 due to the lack of variability in GF_AMS,
but over 95% of the data were well within 10%and all of the marine events data points were well within 10 % of 1:1 line for 110 and 165 nm. Despite the fact that the above regressions suggested a reasonable closure achieved for continental and marine events, the closure results for each individual event turned out to be different. For 75 nm particles in C1 over 60% of estimated GFs were outside 10% range, while in C2 very few GFs fell outside 10% deviation range. In contrast, for 165 nm particles, C1 GFs were typically overestimated while C2 were underestimated. The above results clearly demonstrated an increasing impact
of hydrophobic mode in smaller particles (Figure 4) which was poorly captured by bulk PM1 mass. Marine event GF_HTDMA and GF_AMS were in good agreement, with 95% of data points lying within ±10% of deviation. During individual marine events over 80% of the M1 estimated GFs were underestimated while M2 GFs were mostly overestimated. The regression slope of all the data was 1, indicating that the GF_HTDMA of marine aerosol could be estimated fairly well based on AMS bulk PM1 measurements by using ZSR rule with the sea-salt concentration measured by AMS.
The overall very good agreement was a result of utilising sea-salt mass concentrations derived by AMS (Ovadnevaite et al., 2012). The result verifies AMS as an excellent technique of near real-time sea-salt measurements. During the selected marine events, the PM1 aerosol volume fraction of sea-salt ranged from 2% to 95%. In our study, the use of high time and mass resolution AMS data and subsequent inclusion of sea-salt mass improved the closure greatly. As far as we are aware, this current chemical composition and hygroscopicity closure study is the first conducted on sea-salt containing aerosol.


### 3.4 Closure uncertainty and error analysis

Although a general agreement between measured and estimated GF was found in both continental and marine aerosol, the closure results for each event were slightly different and that motivated us to explore the cause of the closure errors focusing on three metrics: (1) O:C ratio, (2) volume fraction of ammonium nitrate ($vi(NH_4NO_3)$) and (3) aerosol mixing state.
The introduction of a constant $GF_{org}$ was considered to be the cause of a systematic error. The relationship between $GF_{org}$ and organic oxidization level is under intensive debate. In some studies, it was found that $GF_{org}$ increased with increasing O:C ratios in both chamber and ambient studies (Jimenez et al., 2009; Lambe et al., 2011; Massoli et al., 2010; Wong et al., 2011; Wu et al., 2016) and theoretical calculation have demonstrated $GF_{org}$ exhibiting a linear dependence on O:C ratio (Nakao, 2017). However, the correlation between $GF_{org}$ and O:C ratio varied among aerosol sources, and some studies reported no
significant relationship (Chang et al., 2010; Suda et al., 2014).





Fig. 8a shows the relationship between O:C ratio and GF deviation by plotting a normalized error |(GF_HTDMA - GF_AMS) / GF_HTDMA)|. When O:C ratio was below 0.5, a slight overestimation was observed, but the deviation was less than 10%, however, when the O:C ratio was within the 0.5 to 1.25 range no obvious pattern could be discerned. Freshly emitted hydrocarbon compounds have relatively low O:C ratio and lower hygroscopicity, therefore, a $GF_{org} = 1.1$ is likely to cause

overestimation. When the O:C increases to 0.6, the reported $GF_{org}$ ranges from 1.15 to 1.4 depending on the air mass (Hong et al. 2018).

The introduction of a constant density and constant $GF_{org}$, may not be valid for every event, and are likely to cause slight overestimation, but cannot explain the larger deviations above 10%, at least for the aerosol observed at MHD.

Another reason for the apparent overestimation of GF_AMS was considered to be the evaporation of $NH_4NO_3$ in the HTDMA

instrument, which was implicated in previous closure studies causing closure failure during nitrate enriched periods (Gysel et al., 2007; 2001; Swietlicki et al., 1999). Gysel et al. (2007) reported that 50-60% of the volume of $NH_4NO_3$ evaporated within the HTDMA for the particle diameter ranging from 50 to 60 nm. Despite this possible cause of the underestimation of GF in the HTDMA, the presence of nitrate is unlikely to be the main cause for discrepancy in our study for several reasons. First, the residence time of aerosols in our HTDMA system is about 10 s, which is significantly shorter than other systems such as the

HTDMA in Gysel et al. (2007), which has a residence time of approximately 60 s and resulted in significant $NH_4NO_3$ evaporation. Second, no obvious correlation was found between GF deviation and a volume fraction (vi) of $NH_4NO_3$ (R= 0.07, Fig.8 b). Therefore, as far as our study is concerned there is no evidence for ammonium nitrate evaporation responsible for closure discrepancy.

The mixing state of aerosol can influence the hygroscopicity closure in two ways: 1) the accuracy of AMS measurements is

determined by the collection efficiency which for internally mixed aerosols is constant for all the particles, while in external mixtures the application of a constant collection efficiency may produce differences between the real and measured chemical species concentration. In addition to that, externally mixed aerosol have size-dependent chemical composition where bulk chemistry will tend to be more representative of larger sized particles (165 nm) that carry the bulk of mass over smaller sized particles (35 nm) which contribute negligibly to mass. While bulk chemical measurements of PM1 mass may induce errors at

smaller sizes in external mixtures, internally mixed aerosol would likely show little difference between the bulk chemistry and that of smaller sized particles.

Two metrics were adapted to represent the mixing state: (1) GF spread factor and (2) number fraction of NH and LH modes, nf(NH+LH), both of which were derived from the GF-PDF of 165 nm aerosols. The GF spread factor was defined as the standard deviation of GF-PDF divided by an arithmetic mean GF (Stolzenburg and McMurry, 1988). As the MH mode existed

in every event at every particle size, the number fraction of NH and LH (nf(NH+LH)) modes could be used as a metric of the external mixing (Ching et al., 2017; Su et al., 2010). As shown in Fig. 8c and Fig. 8d, the GF spread factor and nf(NH+LH) factor have the largest regression coefficient with GF deviation, the R being 0.51 and 0.57, respectively.

As shown in Fig. 8c, when GF spread factor is less than 0.2, most of GF deviations stay within 10%. When GF spread factor increases over 0.2, the GF deviation starts to increase up to 30%. Similar relationship is found between nf(NH+LH) and GF



deviation, but the tendency is constrained within 10% and many outliers cannot be captured by increasing nf(NH+LH). This is because the relationship between nf(NH+LH) and GF spread factor is not linear. Certain aerosol populations, for example, external mixture of inorganics and sea-salt, that does not contain NH and LH mode, could exhibit a larger GF spread factor. It is also possible that aerosol with a small spread factor value can contain significant number of particles in NH and LH modes. Above all, we conclude that, although the scatter is larger, aerosol with a larger GF spread factor tend to be associated with

high GF deviation. Two examples of GF-PDFs with a GF spread factor larger than 0.2 are shown in Fig. S3. These examples are suggesting the existence of multi-modal distribution. Comparing their chemical composition during C and M events, it was found the presence of sea-salt and elevated BC containing particles, as shown in Fig. S4, suggesting that the externally mixed and anthropogenically impacted aerosol and/or sea-salt containing polluted aerosol are responsible for the discrepancy. Therefore, we suggest that great care must be exercised in estimating hygroscopicity with bulk PM1 chemical composition,

when BC exceeds 0.1 µg m$^{-3}$ and sea-salt is below 0.5 µg m$^{-3}$. It has to be noted that the frequency of occurrence of GF spread factor > 0.2 is as low as 1 % at MHD suggesting that, for most of the time, sea-salt containing bulk PM1 chemical composition can be used to achieve closure with aerosol hygroscopic properties.

## 4. Summary

In this study, data from an HTDMA and an AMS deployed at Mace Head atmospheric research station were used to characterize aerosol hygroscopicity and to elucidate the link with aerosol chemical composition taking the advantage of high temporal resolution of the two instruments. In winter, a period of low biological activity at Mace Head, all sampled aerosols were mostly externally mixed, as revealed by the growth factor probability distribution functions. The continental and marine air masses were examined in detail in terms of the influence of chemical components on aerosol GFs and marine aerosol had

significantly higher hygroscopicity than continental aerosol. A general agreement was achieved between estimated and measured growth factors for 165 nm aerosols. For continental events aerosols, a general agreement was achieved between estimated and measured hygroscopicity GF for $D_0$ 165 nm aerosol, while for marine aerosols, the GF of particles larger than 75 nm could be estimated reasonably. A closure between hygroscopicity and chemical composition was achieved for the first time for marine aerosol with large sea-salt mass loading without significant systematic errors. The ZSR rule for hygroscopicity

estimation of marine aerosol was validated for the first time with sea-salt measured by HR-ToF-AMS.

The analysis on statistical deviations from the perfect closure indicated that the highly external mixing state (GF spread factor > 0.2) can have the largest impact when comparing hygroscopicity derived from bulk chemical composition data and size-dependent hygroscopicity measurements. This study opens up new opportunities for predicting the physico-chemical properties of marine aerosols with HR-ToF-AMS. It should be noted that the good closure for marine aerosol has only been validated for

winter time, and has yet to be explored for summer time when both primary and secondary biogenic organic matter concentrations are expected to be at their highest due to enhanced ocean biological productivity.

**Author contributions**





Colin O'Dowd and Darius Ceburnis conceived the study, Wei Xu analysed the data, Jurgita Ovadnevaite provided the AMS
data, Wei Xu and Kirsten N. Fossum prepared the manuscript with the contribution from all co-authors.

**Acknowledgements**

Authors acknowleged Dr. Jakub Bialek for acquiring HTDMA data. Chinese Scholarship Council (No. 201706310154) is
acknowledged for the fellowship of Mr. Wei Xu. The authors also like to acknowledge EPA-Ireland (AEROSOURCE, 2016-
CCRP-MS-31), the COST Action CA16109 (COLOSSAL) and MaREI (Marine and Renewable Energy Ireland).

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





**Table 1. Density and GF (at 90% RH) of chemical species used in the closure study.**

|  | Density kg m$^{-3}$ | GF |
|---|---|---|
| (NH$_4$)$_2$SO$_4$ | 1769 | 1.71 |
| NH$_4$HSO$_4$ | 1780 | 1.7 |
| H$_2$SO$_4$ | 1830 | 2.05 |
| NH$_4$NO$_3$ | 1720 | 1.81 |
| Sea-salt | 2165 | 2.22 |
| MSA | 1481 | 1.71# |
| Organics | 1400 | 1.18* |
| BC | 1650 | 1 |

#adapted from Fossum et al. 2018.

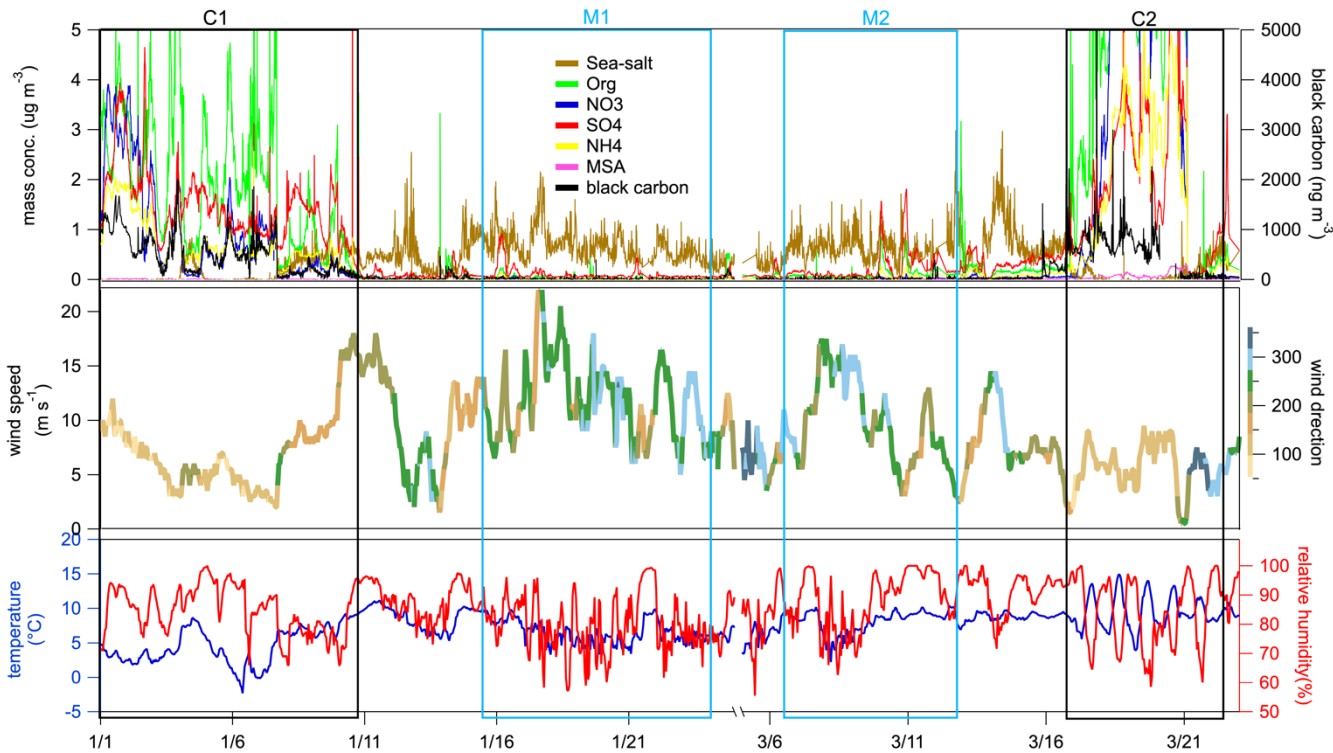

**Figure 1. Temporal variation of mass concentration of chemical species measured by the HR-ToF-AMS and MAAP (top panel);**
**wind speed (m s$^{-1}$) with wind direction represented as a color-scale (middle panel); temperature (ºC, blue line, bottom panel) and**
**RH (hPa, red line, bottom panel). Boxed areas correspond to continental events C1, C2 (in black) and marine events M1, M2 (in**
**blue).**




**Figure 2. 72 h backward trajectories arriving at 500 m above mean sea level at Mace Head retrieved with Global Data Assimilation System for continental events (C1: red dash line, C2: green dash line) and marine events (M1: purple solid line, M2: purple solid line). The trajectory was calculated every 24 hours over the event duration.**





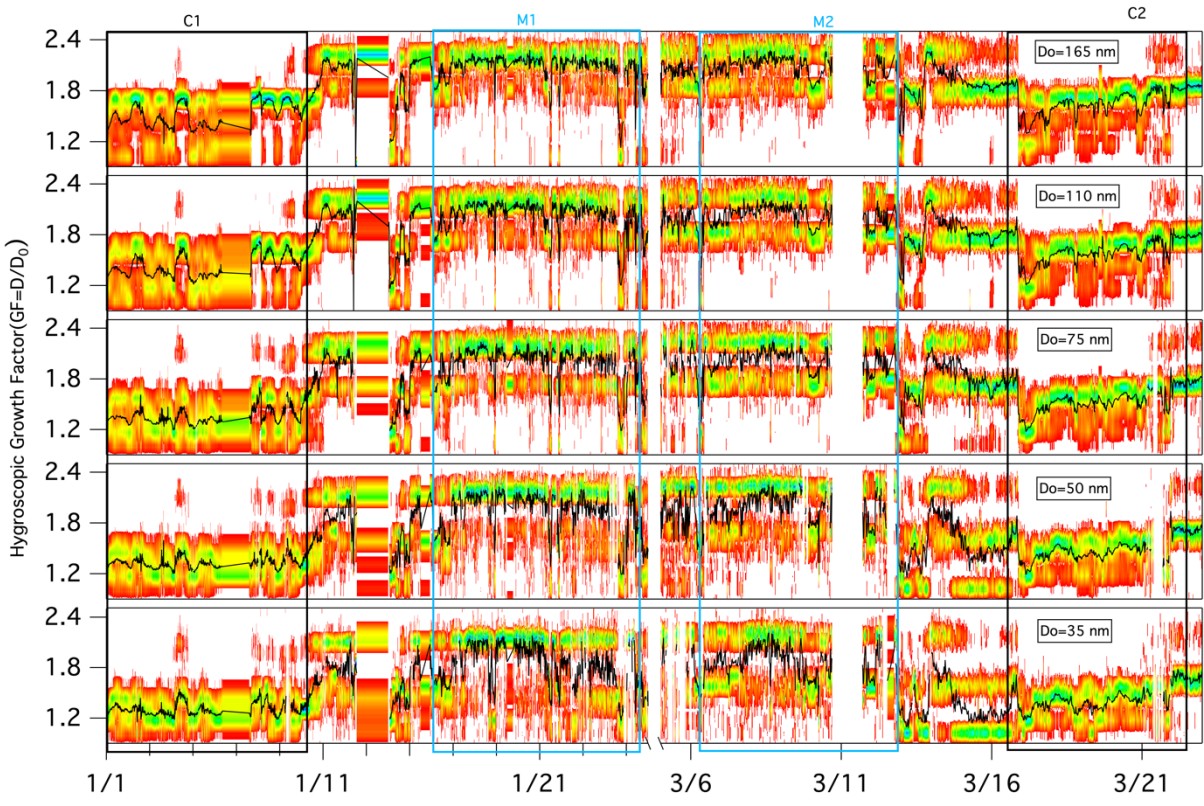

**Figure 3. Growth factor probability distribution function (GF-PDF) for different particle dry sizes as measured by the HTDMA.**
**The colour bar indicates the probability density and the black line represents the averaged GF.**



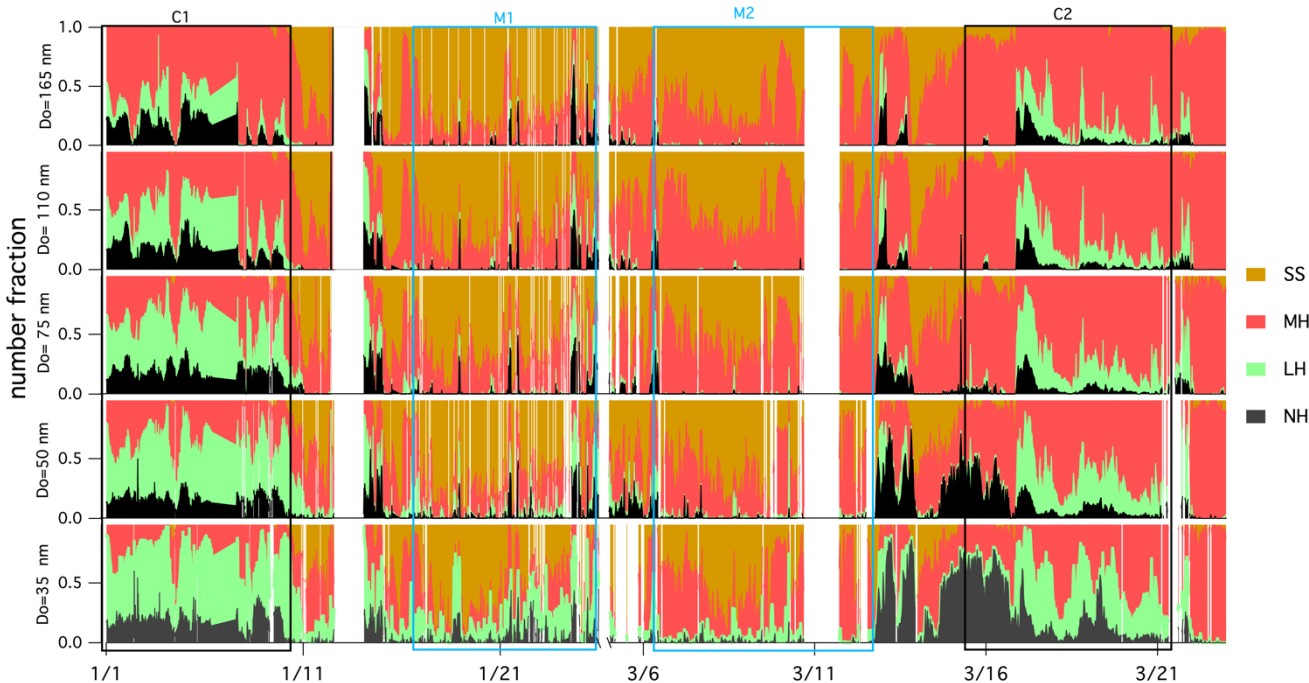

**Figure 4.** Time series of the number fraction of NH mode in black (GF<1.11), LH mode in green, (1.11<GF<1.33), MH mode in red(1.33<GF<1.85) and SS mode in brown (GF>1.85) of aerosols with pre-selected dry diameter.




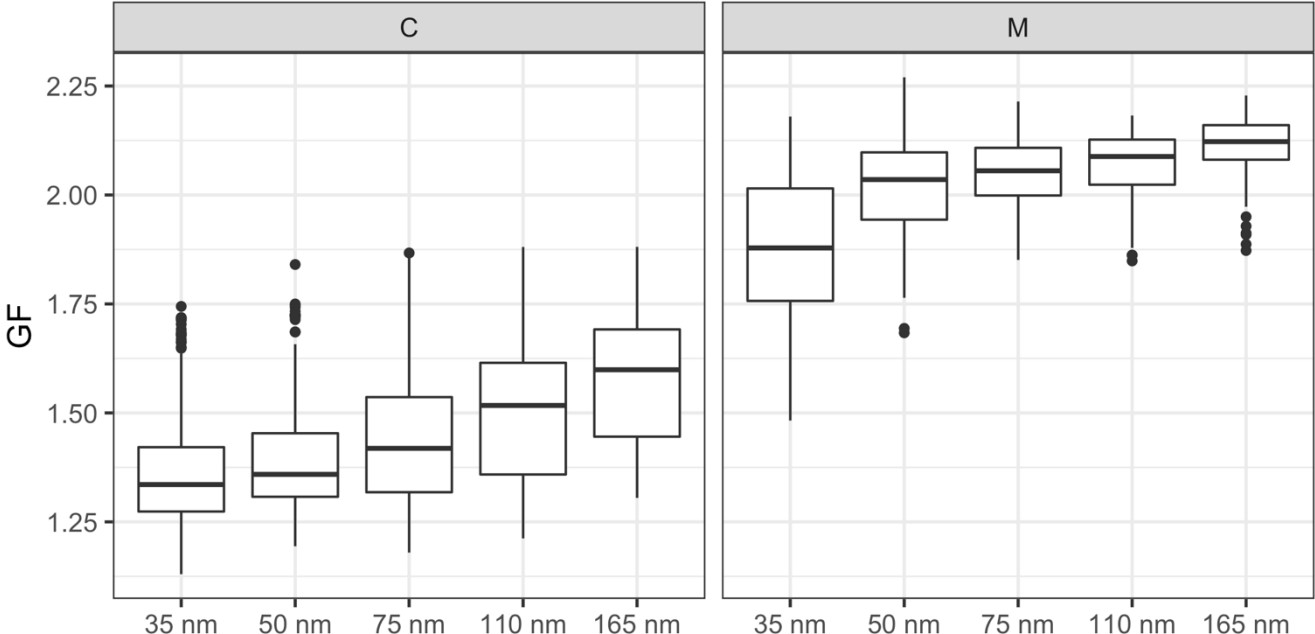

**Figure 5. Size resolved GFs for (a) C1 & C2 and (b) M1 & M2. The horizontal lines represent median GF, the boxes represent 25-75 % percentile and whiskers represent 1.5*IQR from the boxes (where the IQR is the interquartile range). Data beyond the end of whisker are plotted individually as outliers.**






**Figure 6. The comparison of bulk GF_AMS with size dependent GF_HTDMA. The 1:1 line is in black with a 10% deviation indicated by the dashed lines, blue line is the regression line: y=b + a*x, R² is the regression coefficient (variance).**








**Figure 7. The relationship between GF-AMS and GF_HTDMA ($D_0$ = 75, 100 and 165 nm) of separated continental events (a, C1 in red, C2 in green) and marine events (b, M1 in blue, M2 in purple). Black line is 1:1 line for both C or M events, dash lines are 10% deviation and blue line is the regression equation.**


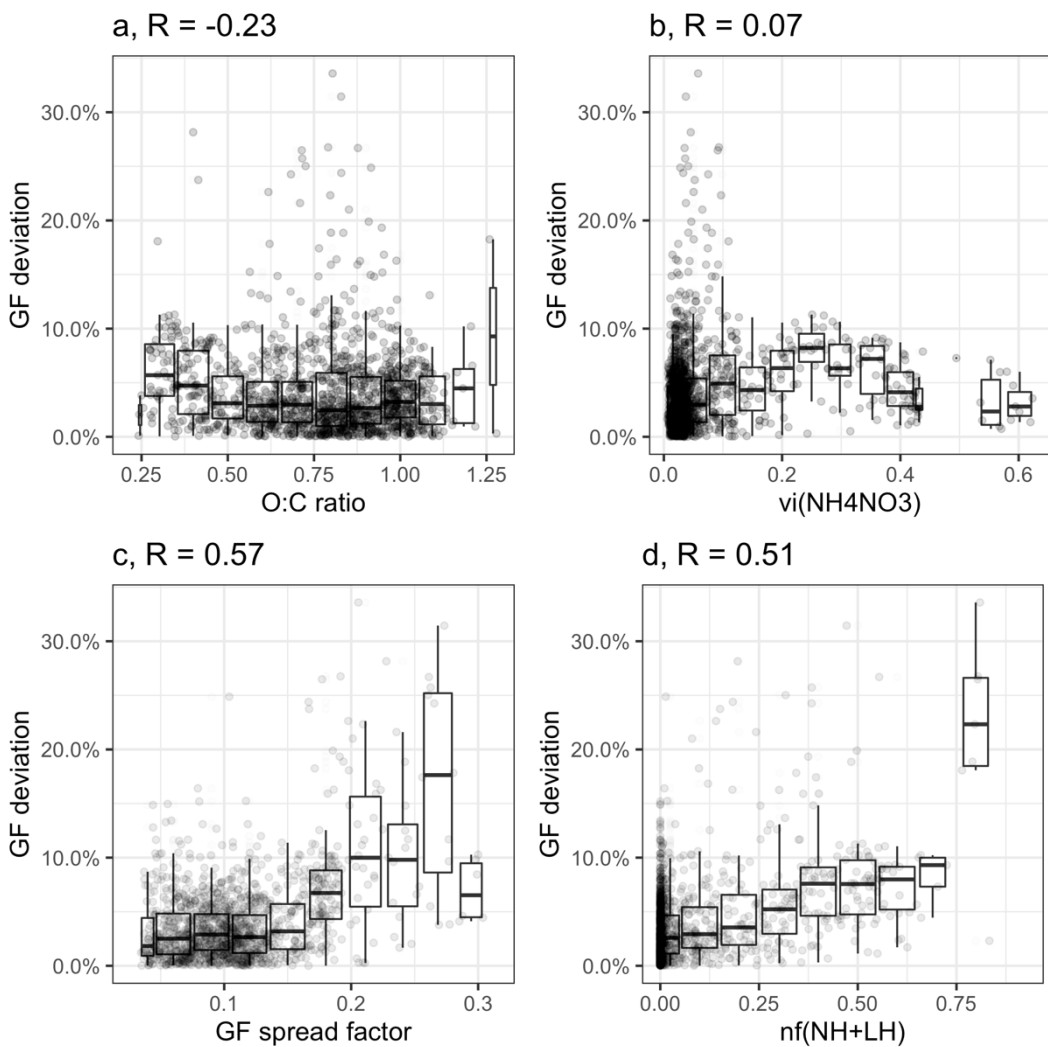

**Figure 8. The relationship of GF deviation to (a) O:C ratio, (b) volume fraction of NH₄NO₃, c) GF spread factor and d) nf(NH+LH) over the whole period.**
