# Peer review of "Aerosol hygroscopicity and its link to chemical composition in coastal atmosphere of Mace Head: marine and continental air masses"

_Atmospheric Chemistry and Physics, 2019_

## Referee Comment (RC1) · Anonymous Referee #1 · 29 Nov 2019

Xu et al. measured size-resolved aerosol hygroscopicity and chemical composition using online techniques at a coastal site (Mace Head) for almost three months in winter, and carried out hygroscopicity-chemistry closure analysis. They found that hygroscopicity showed different diurnal patterns for continental and marine air masses, in general the measured growth factors at 90% RH agreed well with those predicted from aerosol chemical composition. Marine aerosols play a vital role in the climate system, and online and simultaneous measurements of their hygroscopicity and chemical composition are rather limited. Therefore, the results presented are scientifically significant, and the work has been well conducted. I would recommend it for final publication after the following comments (most of which are minor) are addressed.

**Scientific comments:**

Line 37: A recent review paper (Tang et al., A review of experimental techniques for aerosol hygroscopicity studies, Atmos. Chem. Phys., 19, 12631-12686, 2019) summarized what aerosol hygroscopicity is and why it matters, and the authors may consider including it in the revised manuscript.

Line 41-57: More detailed and more insightful discussion on previous work should be provide here. The current manuscript does not convince me in terms of its novelty when compared to previous studies.

Line 41-54: In addition, it is not clear to me why previous AMS measurement could not measure sea salt but the work presented could do so. More details should be given here as well as in Section 2.2.2.

Line 133-134: A recent study (Tang et al., Impacts of methanesulfonate on the cloud condensation nucleation activity of sea salt aerosol, Atmos. Environ., 201, 13-17, 2019.) measured CCN activity of methanesulfonates, and the kappa value of sodium methanesulfonate was determined to be 0.46, giving a GF of 1.72 at 90% RH. This experimental work supports the GF used in this manuscript and should be cited.

Line 240-244: please explain why different size dependence was observed for marine and continental air masses.

Figures 5 and 7: The two figures are a little confusing. I assume "C" means "continental" and "M" means "Marine"? More details should be provided in these two figures and figure

captions. In general I feel that abbreviations have been overused in this manuscript, reducing its readability, and I would suggest that the authors significantly reduce the usage of abbreviations in the revised manuscript.

---

## Referee Comment (RC2) · Anonymous Referee #2 · 22 Jan 2020

Xu et al. report stationary measurements from Mace Head during the winter of 2009. They used HTDMA technique to measure the aerosol hygroscopicity and then compared measurements with hygroscopicity calculations using the aerosol chemical composition (using an HR-ToF-AMS) for size-selected particles. Overall, they found good closure between measured and calculated hygroscopicity assuming complete internal mixture for all components. The manuscript is well written, and the conclusions are justified. I recommend publication after the authors address the following comments.

General comments:

The authors compare measured and calculated hygroscopicity for size-selected particles. However, the chemical composition measured with an AMS for particles smaller than 50 nm cannot be trusted due to significant inlet losses. I suggest that either the authors add adequate justification that the AMS they are using can accurately quantify the mass composition for sub-50nm particles or entirely remove the comparison between calculated and measured hygroscopicity for the sub-50 nm particles.

Specific comments:

Line 100: The authors use the composition-dependent CE (CDCE) to correct AMS concentrations. However, the CDCE method does not take into account sea-salt and organic particles which are ubiquitous in the marine environment. Therefore, I expect that the authors provide a rationale for using this CE correction method. The authors should estimate the CE using a mass closure approach (using DMA volumes and densities) to get another estimate of CE. If the two agree, then this should provide the rationale needed. If not, further discussion is needed.

Line 120: If both sea-salt and sulfuric acid have GF than are larger than 1.85, why call it sea-salt mode? This is confusing. Maybe use highly hygroscopic.

Line 131: Please provide a reference for the choice of density for the organic compounds that is relevant to the marine environment.

Line 160: I do not follow the rationale of only including measurements when BC concentrations were below 15 ng/m3 and then claim that these represent pristinely clean conditions. Surely if continental air masses spent several days above the oceans (while being diluted with cleaner air from the free troposphere) one would expect low BC levels (<15 ng/m3) however, the origin of the particles would still be transported pollution from the continents. Another potential source of pollution could be diluted ship exhaust. I suggest the authors include in the SI a scatter plot showing the non-refractory organic and sulfate concentrations measured by the AMS versus MAAP BC concentrations. If the two are not correlated than this would at least eliminate the influence of combustion aerosols.

Lines 202-203: Can the authors expand as to why does the boundary layer height should affect an aerosol intensive property like hygroscopicity?

Lines 207-208: I am not sure I follow the argument that the authors are trying to make. The authors are claiming that because of the relatively low MSA concentrations measured during continental periods therefore there is low impact from marine sources. However, MSA concentrations during marine periods were clearly lower than those measured during continental periods (Table S2). Also, include details about how MSA concentrations were measured.

Lines 221-223: How do the authors confidently attribute the highly hygroscopic particles observed at 35 nm to sea-salt and not to sulfuric acid? Without providing evidence that these are indeed sea-salt particles, I suggest that the statement be removed.

Line 230: A recent article by Quinn et al. (2019) (https://agupubs.onlinelibrary.wiley.com/doi/full/10.1029/2019JD031740) reported a persistent organic and non-volatile (at 230oC) ultrafine particle mode that was likely entrained from the free troposphere, from measurements in the North Atlantic.

Lines 240-243: I am not sure what is the point that the authors are trying to make by pointing out the different size dependence of the GF during marine and continental periods. Expand or remove.

Line 246: "HTDMA" and not "HTMDA".

Line 224-259: Poorly written and confusing. Please re-write.

Line 263: I am not sure what Fig 7a and 7b refer to. Figure 7 is a six-panel figure and the left and right columns refer to continental and marine periods respectively. Adjust.

Line 264: Adjust the text to: "R2 values were 0.47 and 0.18 for 75nm particles during continental and marine events".

Line 269: perhaps "dynamic range" is better suited than "variability" in this context.

[Figure]

Line 282: I do not agree with the statement that the AMS is an excellent instrument to measure sea-salt. Did the authors collect filters for IC analysis to retrieve sodium and chloride concentrations and then compare these measurements with AMS measurements of sea-salt for this particular study? Or are the authors simply relying on an old calibration from a 2012 study? Other studies in the literature have failed to get good closure between AMS salt measurements and those from IC filters. Can the authors also provide scatter plots of AMS salt concentrations versus wind speed? How do these compare?

Figures S3 and S4 in the SI are mislabeled. Please adjust.

---

## Author Comment (AC1) · 14 Feb 2020

Please see the attached document with the response to both reviewers and a new manuscript version with tracked changes.

Please also note the supplement to this comment:
https://www.atmos-chem-phys-discuss.net/acp-2019-839/acp-2019-839-AC1-supplement.pdf

---

## Author Response (AR2)

We thank the editor for his detailed and constructive review which helped to improve our paper. The review comments are marked **in bold** and followed by our comments and answers and the corresponding changes in the main text is highlighted.

1. **Can you please add some text that answers the general comment of reviewer #2 (about sub-50 nm comparison) in the manuscript as well, instead of simply in the reply to reviewers? That text should clearly support why you believe the AMS data you use can quantitatively measure size composition in that size range.**

The text has been added in section **3.3 chemical composition closure study**

"As described in the method section, the chemical compositions measured by AMS in this study were bulk PM1 chemical composition where contribution of sub-50nm particles was negligible. Consequently, it is expected that the calculated GFs deviates considerably from the measured growth factor for particles smaller than 50 nm."

2. **Can you add some supporting text for the new figures S1 and S8? The reader would see it out of context and won't understand why it is there.**

Some description regarding Figure S1 has been added in methods section:
"The composition dependent collection efficiency (CDCE) was used to correct AMS species concentrations (Middlebrook, Bahreini et al. 2012). CDCE does not take into account sea salt nor organic matter contribution; however, those species would only be corrected proportionally to the total mass (if at all) and would not affect fractional contribution of species which is used in this study. We attempted a comparison between SMPS volume and AMS plus BC combined volume to attest the CDCE correction. The comparison is presented in Figure S1 (the slope was $1.03 \pm 0.01$). The excellent agreement in volume comparison suggested that CDCE correction was realistic with few outliers pointing at slight size range discrepancy between SMPS and AMS."

Reference: Middlebrook, A. N., Bahreini, R., Jimenez, J. L. and Cangaratna M. R.: Evaluation of composition-dependent collection efficiencies for the Aerodyne Aerosol Mass Spectrometer using field data, Aerosol Sci. Tech., 46(3), 258-271, doi: DOI: 10.1080/02786826.2011.620041, 2012.

Some description regarding Figure S8 has been added in discussion section:

"The strict and conservative BC criterion has been used to filter the cleanest maritime air masses. An analysis of the representativeness of clean maritime air masses has been extensively discussed by O'Dowd et al. (2013) where no correlation was found between organic matter (OM) and BC for different BC concentration ranges of 0 -15 ng m$^{-3}$, and 15 -50 ng m$^{-3}$ ($R^2 = 0.006$ and $R^2 = 0.046$, respectively). Figure S8 showing the non-refractory organics and non-sea-salt sulfate concentration by AMS versus BC concentration in this study again demonstrating no relationship in clean marine air masses using conservative BC criterion. However, there was a relationship between the species in continental air masses as one would expect where pollutants are typically

internally mixed and advected by long-range transport to the site."

Some description regarding Figure S8 has been added in discussion section:

3. **I am surprised by the linear-like correlation that appears in figure S2. The source functions of sea salt in models have a much stronger relationship of sea salt with wind speed, usually to the power of around 3. Can you please comment?**

The wind speed and sea salt relationship can be distorted from the one dictated by sea spray source function by a variety of reasons and measurement period. We first attempted a comparison over the three months period producing only tentative relationship. In our second attempt we produced a relationship using data of the entire year and produced a stock chart which now resembles a true source function relationship more closely. The text has been modified accordingly to reflect that.

"The relationship between sea salt and wind speed is presented in Figure S2 for the entire year of 2009 and was similar to the one previously published by Ovadnevaite et al. (2012) although not as clear-cut as previously published. It must be noted that despite the wind speed being the dominant factor for sea salt aerosol production, there were few more parameters in the sea spray source function related to the sea state, salinity and temperature, all affecting the quantitative relationship. The excellent agreement can only be expected in very well-defined low pressure systems producing sea salt events where sea salt is well mixed and filled in the entire boundary layer as it only happens during significant storms. While the sea spray source function is at work during every occasion of wind speed induced bubble bursting, quantitative representation of particle mass and number is not instantly achieved. Nevertheless, both relationships provided extra confidence on the quantitative detection of sea salt by AMS."

[Figure]

Figure S2. Sea salt concentration measured by AMS versus wind speed during 2009, the lines represent medians, the boxes represent 25-75% percentile and whiskers represent 1.5 interquartile range

Reply the reviewer #1

We thank the anonymous reviewer #1 for his/her detailed and constructive review which helped to improve our paper. The review comments are marked **in bold** and followed by our comments and answers and the corresponding changes in the main text is highlighted.

**Xu et al. measured size-resolved aerosol hygroscopicity and chemical composition using online techniques at a coastal site (Mace Head) for almost three months in winter, and carried out hygroscopicity-chemistry closure analysis. They found that hygroscopicity showed different diurnal patterns for continental and marine air masses, in general the measured growth factors at 90% RH agreed well with those predicted from aerosol chemical composition. Marine aerosols play a vital role in the climate system, and online and simultaneous measurements of their hygroscopicity and chemical composition are rather limited. Therefore, the results presented are scientifically significant, and the work has been well conducted. I would recommend it for final publication after the following comments (most of which are minor) are addressed.**

Thank you for the comment.

**Scientific comments:**

**Line 37: A recent review paper (Tang et al., A review of experimental techniques for aerosol hygroscopicity studies, Atmos. Chem. Phys., 19, 12631-12686, 2019) summarized what aerosol hygroscopicity is and why it matters, and the authors may consider including it in the revised manuscript.**

The review paper has been added to the list of references.

The hygroscopic growth factor of aerosol particles was measured with an HTDMA (Liu et al., 1978; Rader and McMurry, 1986; Swietlicki et al., 2008; Tang et al., 2019)

Tang, M. J., Chan, C. K., Li, Y. J., Su, H., Ma, Q. X., Wu, Z. J., Zhang, G. H., Wang, Z., Ge, M. F., Hu, M., He, H., and Wang, X. M.: A review of experimental techniques for aerosol hygroscopicity studies, Atmos. Chem. Phys., 19, 12631-12686, doi: 10.5194/acp-19-12631-2019, 2019.

**Line 41-57: More detailed and more insightful discussion on previous work should be provide here. The current manuscript does not convince me in terms of its novelty when compared to previous studies.**

We have added more discussion of the previous studies:

"For example, closure study conducted in Paris revealed an over-estimation of predicted hygroscopicity when nitrate mass concentration exceeded 10 μg m$^{-3}$ (Kamilli et al., 2014). A closure study in Beijing suggested that the hygroscopicity of organics was related to their

oxidized state (Wu et al., 2016), while another study in Hongkong did not find any improvement in closure (Yeung et al., 2014). Despite the advantage of collocated aerosol chemical composition and hygroscopicity measurements helping to reconcile sub-saturated particle hygroscopicity with its chemical composition thereby identifying knowledge gaps, it is widely accepted that sea-salt (the main component of marine aerosol) measurements by AMS are challenging because of its semi-refractory nature resulting in incomplete chemical composition and unrealistic hygroscopicity."

Kamilli1, K. A., Poulain, L., Held, A., Nowak, A., Birmili, W. and Wiedensohler, A.: Hygroscopic properties of the Paris urban aerosol in relation to its chemical composition, Atmos. Chem. Phys., 14(2), 737–749, doi:10.5194/acp-14-737-2014, 2014.

**Line 41-54: In addition, it is not clear to me why previous AMS measurement could not measure sea salt but the work presented could do so. More details should be given here as well as in Section 2.2.2.**

The following text has been added in section 2.2.2 :

"The AMS typically runs at evaporation temperature of 600 °C, which is optimized for the detection of non-refractory aerosol species such as organic matter, nitrate, sulfate and ammonium. Sea salt was expected to be refractory at the above temperature and the quantification by AMS could only be realized at higher temperatures thereby compromising detection of non-refractory species (Allan et al., 2004). However, Ovadnevaite et al. (2012) has convincingly demonstrated that sea salt can be successfully quantified at the standard evaporation temperature as long as relative humidity is maintained within reasonable limits (< 80%) and the AMS vaporizer is not overloaded by sea salt."

**Line 133-134: A recent study (Tang et al., Impacts of methanesulfonate on the cloud condensation nucleation activity of sea salt aerosol, Atmos. Environ., 201, 13-17, 2019.) measured CCN activity of methanesulfonates, and the kappa value of sodium methanesulfonate was determined to be 0.46, giving a GF of 1.72 at 90% RH. This experimental work supports the GF used in this manuscript and should be cited.**

The suggested study has been added to references.

"The $GF_{MSA}=1.71$ was calculated by kappa value which in turn was obtained by AIOMFAC model (Fossum et al., 2018; Zuend et al., 2011) and supported by a recent lab experiment (Tang et al., 2018)."

Tang, M., Guo, L., Bai, Y., Huang, R.-J., Wu, Z., Wang, Z., Zhang, G., Ding, X., Hu, M., Wang, X.: Impacts of methanesulfonate on the cloud condensation nucleation activity of sea salt aerosol. Atmos. Environ., 201:13-17, doi: 10.1016/j.atmosenv.2018.12.034, 2018.

**Line 240-244: please explain why different size dependence was observed for marine and continental air masses.**

The following text has been added to explain the size dependence.

"The size dependence of GF can result from Kelvin effect and (or) chemical composition. To remove the effect of Kelvin effect, the hygroscopicity parameter kappa was calculated. Similar to GFs, the kappa values shown size dependence for both continental and marine events (Fig. S5) The difference size dependence behavior was the result of different air mass history and corresponding aerosol production mechanisms affecting aerosol chemical composition. Marine aerosols are mainly produced by wind stress induced bubble bursting while continental anthropogenic aerosol underwent significant ageing process."

[Figure]

Figure S5. Size resolved kappa values for (a) Continental (C) and (b) Marine (M). The horizontal lines represent median GF, the boxes represent 25-75 % percentile and whiskers represent 1.5*IQR from the boxes (where the IQR is the interquartile range). Data beyond the end of whisker are plotted individually as outliers.

**Figures 5 and 7: The two figures are a little confusing. I assume "C" means "continental" and "M" means "Marine"? More details should be provided in these two figures and figure captions. In general I feel that abbreviations have been overused in this manuscript, reducing its readability, and I would suggest that the authors significantly reduce the usage of abbreviations in the revised manuscript.**

The Figure captions were modified to accordingly:

Figure 5. Size resolved GFs for (a) Continental (C) and (b) Marine (M). The horizontal lines represent median GF, the boxes represent 25-75 % percentile and whiskers represent 1.5*IQR from the boxes (where the IQR is the interquartile range). Data beyond the end of whisker are plotted individually as outliers.

Reply to the reviewer #2 comments

We thank the anonymous reviewer #2 for his/her detailed and constructive review which helped to improve our paper. The reviewer comments are highlighted **in bold** and followed by our answers, the change in main text is highlighted.

**General comments: The authors compare measured and calculated hygroscopicity for size-selected particles. However, the chemical composition measured with an AMS for particles smaller than 50 nm cannot be trusted due to significant inlet losses. I suggest that either the authors add adequate justification that the AMS they are using can accurately quantify the mass composition for sub-50nm particles or entirely remove the comparison between calculated and measured hygroscopicity for the sub-50 nm particles.**

As described in the method section, the chemical compositions measured by AMS in this study were not size-resolved, instead we used bulk PM1 chemical composition where contribution of sub-50nm particles was negligible. Therefore, it is expected that the calculated growth factor deviates considerably from the measured growth factor for particles smaller than 50 nm. As a matter of fact, our HR-ToF-AMS was indeed capable of measuring sub-50nm particles as was proven and corroborated by detailed comparison and sensitivity analysis by Ovadnevaite et al., 2017, doi:10.1038/nature22806.
We would like to keep the comparison for the sub-50 nm particles.

**Specific comments:**
  **Line 100: The authors use the composition-dependent CE (CDCE) to correct AMS concentrations. However, the CDCE method does not take into account sea-salt and organic particles which are ubiquitous in the marine environment. Therefore, I expect that the authors provide a rationale for using this CE correction method. The authors should estimate the CE using a mass closure approach (using DMA volumes and densities) to get another estimate of CE. If the two agree, then this should provide the rationale needed. If not, further discussion is needed.**

The reviewer is correct in saying that CDCE is not taking into account sea salt and organic matter contributions. However, in this paper we used only fractional contributions of chemical species which were independent of collection efficiency. The AMS derived volume and SMPS volume have shown excellent agreement (correlation $R^2 = 0.91$) with very few outliers attributed to the impact of larger particles typically not measured by SMPS (>500 nm).

The following text has been added in Supporting information:

[Figure]

Figure S1. PM1 volume vs SMPS volume. The estimated PM1 volume was calculated by using a species dependent density of 1.40 g cm$^{-3}$ for Org, 1.78 g cm$^{-3}$ for sulfate, 1.72 g cm$^{-3}$ for nitrate, 1.75 g cm$^{-3}$ for ammonium, 1.4 g cm$^{-3}$ for chloride, 1.65 g/cm$^{3}$ for BC, 2.17 g cm-3 for sea salt and 1.48 for MSA.

**Line 120: If both sea-salt and sulfuric acid have GF than are larger than 1.85, why call it sea-salt mode? This is confusing. Maybe use highly hygroscopic.**

The terminology of sea-salt mode and more-hygroscopic mode was followed from Swietlicki et al. (2008). Therefore we would like to keep the term "sea-salt mode" while discussing sulphuric acid contribution where appropriate.

**Line 131: Please provide a reference for the choice of density for the organic compounds that is relevant to the marine environment.**

The density of organics was chosen to represent oxidized organics (e.g. carboxylic acids which are ubiquitous to marine environment) and a reference has already been provided as from Gysel et al. 2007.

"In this study, we first used a fixed GF value of 1.18 for organics which was the averaged value from several closure studies (Wang et al., 2018; Yeung et al., 2014) and a constant density of 1400 kg m$^{-3}$ used by Gysel et al.( 2007)."

**Line 160: I do not follow the rationale of only including measurements when BC concentrations were below 15 ng/m3 and then claim that these represent pristinely clean conditions. Surely if continental air masses spent several days above the oceans (while being diluted with cleaner air from the free troposphere) one would expect low BC levels (<15 ng/m3) however, the origin of the particles would still be transported pollution from the continents. Another potential source of pollution could be diluted ship exhaust. I suggest the authors include in the SI a scatter plot showing the non-refractory organic and sulfate concentrations measured by the AMS versus MAAP BC concentrations. If the two are not correlated than this would at least eliminate the influence of combustion aerosols.**

The strict and conservative BC criterion has been used to filter the cleanest maritime air masses. An analysis of the representativeness of clean maritime air masses has been extensively discussed by O'Dowd et al. (2013) where no correlation was found between organic matter (OM) and BC for different BC concentration ranges of 0 -15 ng m$^{-3}$, and 15 - 50 ng m$^{-3}$ ($R^2$ = 0.006 and $R^2$ = 0.046, respectively). A scatter plot showing the non-refractory organics and sulfate concentration by AMS versus BC concentration in this study were added in the SI again demonstrating no relationship.

As shown in Figure S3, the near-hydrophobic (NH) particles in accumulation mode (110 nm and 165 nm) were removed by applying the clean sector criterion (BC < 15 ng/m3). The remaining Aitken mode NH particles were unlikely coming from transported pollution as being not accompanied by accumulation mode particles of (100-200nm).

[Figure]

Figure S8. The relationship between Org, SO4 versus BC for Continental (C) and Marine (M) case events.

**Lines 202-203: Can the authors expand as to why does the boundary layer height should affect an aerosol intensive property like hygroscopicity?**

The following text was added to increase clarity:
"When the sun rises in the morning, the boundary layer increases in height and the older particles are mixed down. In general, the old particles are more hygroscopic resulting from the cloud processing and photo-aging (Rissler et al., 2006)."

Rissler, J., Vestin, A., Swietlicki, E., G. Fisch, G., Zhou, J., Artaxo, P., and Andreae. M. O.: Size distribution and hygroscopic properties of aerosol particles from dry-season biomass burning in Amazonia, Atmos. Chem. Phys., 6(2), 471–491, doi: 1680-7324/acp/2006-6-471, 2006.

**Lines 207-208: I am not sure I follow the argument that the authors are trying to make. The authors are claiming that because of the relatively low MSA concentrations measured during continental periods therefore there is low impact from marine sources. However, MSA concentrations during marine periods were clearly lower than those measured during continental periods (Table S2). Also, include details about how MSA concentrations were measured.**

We have removed the description of the MSA concentration and the details of MSA measurement have been added in the Methods section. Details about MSA quantification were added with the corresponding reference.

"The quantification of MSA was realized and calibrated by ion signal $CH_3SO_2^+$ and $CH_3SO_3H^+$ which are the exclusively related to MSA. The operational details of the HR-ToF-AMS are described by Ovadnevaite et al. (2014)"

**Lines 221-223: How do the authors confidently attribute the highly hygroscopic particles observed at 35 nm to sea-salt and not to sulfuric acid? Without providing evidence that these are indeed sea-salt particles, I suggest that the statement be removed.**

The following text was added to corroborate our argument in line 250.

"Although the SS mode observed at 35 nm could have been attributed to sulfuric acid, it was unlikely to be the case for the following reasons: (1) the number of SS mode particles (number fraction of SS mode times the number of Aitken mode particles measured by scanning mobility particle sizer) was highly related to wind speed and (2) the ammonium tends to react with smaller sulfate particles because of their larger surface to volume ratio producing less hygroscopic ammonium (bi)sulfate."

**Line 230: A recent article by Quinn et al. (2019) (https://agupubs.onlinelibrary.wiley.com/doi/full/10.1029/2019JD031740) reported a persistent organic and non-volatile (at 230oC) ultrafine particle mode that was likely entrained from the free troposphere, from measurements in the North Atlantic.**

We do not see a direct linkage with Quinn et al. paper because their study did not consider strict filtering method by BC. We would like to note that those NH 35 nm particles are very few and we need longer time periods analyzed during different seasons to unravel their source which can be either due to entrainment or of biogenic origin. We prefer to report the finding of those NH 35 nm particles, but more efforts are required to make a conclusive statement.

**Lines 240-243: I am not sure what is the point that the authors are trying to make by pointing out the different size dependence of the GF during marine and continental periods. Expand or remove.**

The following text has been added to make our discussion clearer:

"The size dependence of GF could be a result of Kelvin effect and chemical composition. To remove the effect of Kelvin effect, the hygroscopicity parameter kappa was calculated. Similar to GFs, the kappa values shown size dependent for both continental and marine events. The remaining size dependent behavior was a result of the different air mass history and corresponding aerosol production mechanism. Marine aerosols produced by wind stress induced bubble bursting, while continental anthropogenic aerosol underwent significant aging process."

**Line 246: "HTDMA" and not "HTMDA".**

Corrected.

**Line 224-259: Poorly written and confusing. Please re-write.**

[revised manuscript text omitted]

**Line 263: I am not sure what Fig 7a and 7b refer to. Figure 7 is a six-panel figure and the left and right columns refer to continental and marine periods respectively. Adjust.**

Adjusted.

"Comparison between GF_AMS and GF_HTDMA was plotted for continental and marine events, as shown in Fig. 7."

**Line 264: Adjust the text to: "R2 values were 0.47 and 0.18 for 75nm particles during continental and marine events".**

Adjusted.

"The regression lines were approaching the 1:1 line with the increasing size. For example, $R^2$ values were 0.47 and 0.18 for 75 nm particles during continental and marine events."

**Line 269: perhaps "dynamic range" is better suited than "variability" in this context**

Modified.

**Line 282: I do not agree with the statement that the AMS is an excellent instrument to measure sea-salt. Did the authors collect filters for IC analysis to retrieve sodium and**

**chloride concentrations and then compare these measurements with AMS measurements of sea-salt for this particular study? Or are the authors simply relying on an old calibration from a 2012 study? Other studies in the literature have failed to get good closure between AMS salt measurements and those from IC filters. Can the authors also provide scatter plots of AMS salt concentrations versus wind speed? How do these compare?**

We have changed the word 'excellent' to 'good'. IC measurements although frequently used, including ourselves, are of only supporting value given very different size ranges corresponding to AMS and IC respectively. Indeed, we used the scaling factor of 51 for quantitative measurement of sea salt form our previous study (Ovadnevaite et al., 2012). The scatter plot of AMS sea salt concentration versus wind speed is given in Figure S1 which demonstrates a good agreement. An excellent agreement can only be expected in very well defined sea salt events where sea salt is well mixed in the entire boundary layer as it happens during significant storms (Ovadnevaite et al. 2012)

The following text has been added to the method section:
"A comparison between SMPS volume and AMS total volume is presented in Figure S1 and the relationship between wind speed and sea salt presented in Figure S2. Both relationships provided extra confidence on the quantitative detection of sea salt by AMS."

[Figure]

Figure S1. PM1 volume derived from AMS vs SMPS volume. The calculated PM1 volume was obtained by using a species-dependent density of 1.40 g cm$^{-3}$ for Org, 1.78 g cm$^{-3}$ for sulfate, 1.72 g cm$^{-3}$ for nitrate, 1.75 g cm$^{-3}$ for ammonium, 1.4 g cm$^{-3}$ for chloride, 1.65 g/cm$^{3}$ for BC, 2.17 g cm-3 for sea salt and 1.48 for MSA.

[Figure]

Figure S2. Sea salt concentration measured by AMS versus wind speed.

**Figures S3 and S4 in the SI are mislabeled. Please adjust.**

Corrected.

[revised manuscript text omitted]